# Polymorphisms in Genes Encoding VDR, CALCR and Antioxidant Enzymes as Predictors of Bone Tissue Condition in Young, Healthy Men

**DOI:** 10.3390/ijms24043373

**Published:** 2023-02-08

**Authors:** Ewa Jówko, Barbara Długołęcka, Igor Cieśliński, Jadwiga Kotowska

**Affiliations:** 1Department of Physiology and Biochemistry, Faculty of Physical Education and Health in Biała Podlaska, Józef Piłsudski University of Physical Education in Warsaw, 00-968 Warsaw, Poland; 2Department of Sports and Training Sciences, Faculty of Physical Education and Health in Biała Podlaska, Józef Piłsudski University of Physical Education in Warsaw, 00-968 Warsaw, Poland

**Keywords:** athletes, bone mineral density, bone mineral content, osteoporosis, SOD2, GPx

## Abstract

The aim of the study was to assess significant predictors of bone mineral content (BMC) and bone mineral density (BMD) in a group of young, healthy men at the time of reaching peak bone mass. Regression analyses showed that age, BMI and practicing combat sports and team sports at a competitive level (trained vs. untrained group; TR vs. CON, respectively) were positive predictors of BMD/BMC values at various skeletal sites. In addition, genetic polymorphisms were among the predictors. In the whole population studied, at almost all measured skeletal sites, the *SOD2* AG genotype proved to be a negative predictor of BMC, while the *VDR FokI* GG genotype was a negative predictor of BMD. In contrast, the *CALCR* AG genotype was a positive predictor of arm BMD. ANOVA analyses showed that, regarding *SOD2* polymorphism, the TR group was responsible for the significant intergenotypic differences in BMC that were observed in the whole study population (i.e., lower BMC values of leg, trunk and whole body were observed in AG TR compared to AA TR). On the other hand, higher BMC at L1–L4 was observed in the *SOD2* GG genotype of the TR group compared to in the same genotype of the CON group. For the *FokI* polymorphism, BMD at L1–L4 was higher in AG TR than in AG CON. In turn, the *CALCR* AA genotype in the TR group had higher arm BMD compared to the same genotype in the CON group. In conclusion, *SOD2*, *VDR FokI* and *CALCR* polymorphisms seem to affect the association of BMC/BMD values with training status. In general, at least within the *VDR FokI* and *CALCR* polymorphisms, less favorable genotypes in terms of BMD (i.e., *FokI* AG and *CALCR* AA) appear to be associated with a greater BMD response to sports training. This suggests that, in healthy men during the period of bone mass formation, sports training (combat and team sports) may attenuate the negative impact of genetic factors on bone tissue condition, possibly reducing the risk of osteoporosis in later age.

## 1. Introduction

Osteoporosis is a common metabolic bone disease with reduced BMD and microarchitectural deterioration of bone tissue, which leads to an increase in bone fragility and fracture risk. This disease is not limited to postmenopausal women. There is increasing attention being paid to osteoporosis in older men. Despite a lower prevalence of osteoporosis in men than in women, men have higher morbidity and mortality rates after fracture [1].

As a multifactorial disease, osteoporosis is influenced by various environmental and genetic factors. One of the general strategies for making the skeleton more resistant to fracture is to maximize peak bone mass by the age of 30, since bone formation is predominant until the age of 25 [2]. Among environmental factors, physical activity habits play a pivotal role in bone mass and structure characteristics [3]. Even in a population of physically active young adult men, with a BMD at the upper end of the normal range, higher levels of physical activity were associated with higher bone mass [4].

Physical activity promotes changes in the bone metabolism through a direct effect (via mechanical force) or an indirect effect (promoted by hormonal factors) [5,6]. Mechanical loadings, including compression, strain and fluid shear, are the stimuli playing essential roles in osteoblast differentiation and mineralization, as well as in maintaining proper high bone mass and density [7]. It can also be assumed that particularly dynamic, weight-bearing sports with short, high and multidimensional loads have strong effects on bone formation, independent of training quantity [8,9].

Apart from the type of physical activity, a proper diet with an adequate intake of calcium and vitamin D is a prerequisite for achieving the highest possible peak bone mass. On the other hand, absorption and metabolism of these dietary compounds, and, in turn, their influence on BMD, varies depending on genetic factors. The vitamin D receptor (*VDR*) gene polymorphisms *ApaI, BsmI, TaqI* and *FokI* are most frequently studied as potential factors predicting osteoporosis risk in middle-aged and older populations [10,11,12]. Additionally, the receptor of calcitonin (*CALCR)* gene polymorphism (C1377T *AluI*)*,* related to calcium and phosphate homeostasis, was found to affect BMD [13], with ambiguous results reported in studies on men from European countries [14,15]. Moreover, associations between collagen type I (*COLIA1)* gene polymorphism (+1245G/T) and BMD were reported in postmenopausal women [16].

Although physical activity may modify the relationship between genetics and bone mineralization, few studies have evaluated the association between *VDR* polymorphisms and BMD in athletes, with limited numbers of participants and conflicting results [17,18]. Furthermore, no data are available regarding the importance of *CALCR* or *COLlA1* gene polymorphisms as potential predictors of BMD in young athletes in the phase of peak bone mass development.

It has also been suggested that dynamic formation and resorption of bone tissue throughout a person’s life may depend on nonenzymatic and enzymatic antioxidant potential (superoxide dismutase, various peroxidases, catalase) [19,20]. Animal studies have shown a positive correlation between BMD and superoxide dismutase (SOD) activity in the blood [21]. In turn, lower blood glutathione peroxidase (GPx) activity was observed in a group of older women and men with osteoporosis compared to in healthy people in the same age range [22].

It is known that enzymatic antioxidant potential can be determined by genetic factors, such as polymorphism in genes encoding enzymatic proteins [23,24]. Some data in the literature indicate that polymorphism of genes encoding antioxidant enzymes (among others, *SOD2 Ala-9Val* and *GPx Pro198Leu*) may affect BMD in the middle-aged and elderly population [20,25,26]. However, there are no reports of potential links between the polymorphisms mentioned above and BMD or BMC in the younger population in the available literature.

Finally, exercise training is suggested to modulate the expression of genes encoding antioxidant enzymes in a positive manner, thus, resulting in an increase in enzymatic antioxidant potential [27]. Therefore, it is interesting to evaluate whether common polymorphisms in genes encoding antioxidant enzymes can predict BMD values in healthy young men. Furthermore, the next question is whether these polymorphisms can modulate the response of bone tissue formation (BMD, BMC values) to high-impact exercise training in professional athletes.

Therefore, the aim of the study was to assess which factors are the most potent predictors of BMC and BMD values in a group of young, healthy men during the period of reaching peak bone mass. The factors taken into account were (1) trained vs. untrained status; (2) anthropometric variables: age, BMI; (3) the gene polymorphisms mentioned above: *VDR* (*ApaI,* rs 7975232; *BsmI,* rs1544410; *FokI* rs 2228570), *COLIA1* (rs 1800012), *CALCR* (rs 1801197) and antioxidant enzymes (*SOD1,* rs2234694; *SOD2,* rs4880; and *GPx* rs1050450); (4) blood levels of bone metabolism markers: calcium, phosphates, vitamin D (25-OH D), osteocalcin, alkaline phosphatase; (5) blood prooxidant-antioxidant homeostasis: total antioxidant capacity, uric acid, lipid hydroperoxides, superoxide dismutase, glutathione peroxides.

## 2. Results

The anthropometric characteristics of participants in the CON and TR groups are shown in Table 1. No intergroup differences were found in these parameters (*p* > 0.05).

Individuals from the CON and TR groups differed significantly in BMC value of the lumbar spine (*p* = 0.00004), whereas the groups did not differ in BMC of other points of the skeleton (Table 1). In turn, significantly higher BMD values (lumbar spine, *p* = 0.0008; arm, *p* = 0.03; trunk, *p* = 0.025) were seen in the TR group as compared to the CON group (Table 1). This significant difference was also apparent for the Z-score (*p* = 0.027), whereas only an increasing trend in the T-score (*p* = 0.076) was observed in the TR group as compared to CON group (Table 1). None of our participants had an abnormal T-score, i.e., below −1.

Table 2 presents the level of biochemical parameters in the blood (with reference range for population studied). In the TR group, significantly lower plasma TAC levels (*p* = 0.002), as well as plasma UA concentration (*p* = 0.0001), were shown in comparison to in the CON group (Table 2). No significant differences were observed between the groups in the level of other parameters in the blood. In both groups (TR and CON), the mean concentration of 25-OH D in serum was within the reference values (30–50 ng/mL). We also measured the correlations between biochemical parameters (data are not shown). In the whole study population, the only significant correlation was seen between TAC and UA (r = 0.54, *p* < 0.00001). Furthermore, when all participants were divided into two subgroups, taking into account individual vitamin D status (serum 20-OH D), a negative correlation between serum 25-OH D and OC was observed in the subgroup with 25-OH D < 30 ng/mL (r = −0.32; *p* < 0.01) but not in the subgroup with 25-OH D ≥ 30 ng/mL.

Genotype frequencies for subjects in the CON and TR groups are shown in Table 3. In both the CON and TR group, the genotype frequencies within *GPx* Pro198Leu polymorphism differed significantly (*p* = 0.0008 and *p* = 0.005, respectively) from those expected under the Hardy–Weinberg equilibrium, with the TT genotype being absent in our study population (and no differences between the CON and TR groups). In the case of *CALCR* polymorphism, it was revealed that, in the TR group (but not the CON group), the genotype distribution did not follow the Hardy–Weinberg distribution (*p* = 0.00006). Moreover, significant differences in genotype frequencies within *CALCR* polymorphism were found between groups, with GG being more frequent in the TR than the CON group (*p* = 0.003). For other SNPs in the genes analyzed in our study, the χ^2^ test confirmed that the observed frequencies did not deviate from the Hardy–Weinberg equilibrium, with no significant intergroup differences in genotype frequencies (Table 3).

The predictors of BMC and BMD, as evaluated by regression models, are presented in Table 4 and Table 5, respectively. Age was a positive predictor of BMC at two points of the skeleton, L1–L4 (*p* < 0.01) and arm (*p* < 0.05), whereas BMI positively affected BMC in almost all the analyzed areas (*p* < 0.001), except for in L1–L4 (Table 4). Although the status “TR vs. CON” (i.e., trained vs. untrained) was not a predictor of BMC values (Table 4), all kinds of sports disciplines studied (i.e., MMA, soccer, handball and wrestling) positively affected the BMC value at L1–L4 (*p* < 0.05 for MMA and soccer; *p* < 0.001 for handball and wrestling). In turn, handball was the positive predictor of BMC in all measured body parts (*p* < 0.001). Moreover, two polymorphisms of genes related to enzymatic antioxidant defense were factors influencing BMC (Table 4). Within *GPx* Pro198Leu polymorphism, genotype CT was a negative predictor of BMC at L1–L4 (*p* < 0.05); however, ANOVA analyses did not reveal any differences in BMC values between CC and CT genotypes either in the whole study population or in the TR and CON groups. In turn, amongst the three genotypes of *SOD2* Ala-9 Val polymorphism, AG was the independent factor that had a negative impact on BMC values in all body parts except the arm (*p* < 0.05), with GG being a negative predictor in non-significant manner (Table 4). 

In the case of BMC at L1–L4 in the whole studied group of subjects (both control and trained), a significantly lower value (*p* = 0.03) was found in the SOD2 AG genotype as compared to in genotypes AA and GG (Figure 1A; with main effect of *SOD2*, *p* = 0.02). Moreover, when this parameter was analyzed in the CON and TR groups, BMC at L1–L4 was significantly higher in GG TR than in GG CON (*p* = 0.02; Figure 1B; with main effect of *SOD2*, *p* = 0.01 and TR vs. CON, *p* = 0.000000001).

Regarding the BMC of the leg (Figure 2A), trunk (Figure 3A) and whole body (Figure 4A), when analyzed in all the participants together (from both the CON and TR groups), the AG genotype had lower values in comparison with the AA genotype (leg: *p* = 0.006, main effect of *SOD2*, *p* = 0.008; trunk: *p* = 0.002, main effect of *SOD2*, *p* = 0.002; whole body: *p* = 0.003, main effect of *SOD2*, *p* = 0.004). After dividing the whole studied group into the TR and CON groups, the BMC of the leg (Figure 2B), trunk (Figure 3B) and whole body (Figure 4B) was lower in AG TR than AA TR (leg: *p* = 0.03, main effect of *SOD2*, *p* = 0.008; trunk: *p* = 0.025, main effect of *SOD2*, *p* = 0.002; whole body: *p* = 0.057, main effect of *SOD2*, *p* = 0.004).

As shown in Table 5, the factor that affected BMD to the greatest extent was practicing sports (TR vs. CON status), which concerned BMD at L1–L4 (*p* < 0.001) and the arm (*p* < 0.01). A positive impact of MMA on the arm’s BMD value was found (*p* < 0.05). Wrestling had a positive impact on the BMD of the arm and trunk (*p* < 0.001 and *p* < 0.05; respectively), whereas handball positively affected the BMD of the arm, leg, trunk and total body (*p* < 0.001). 

Similar to BMC, the subsequent predictors of BMD were age and BMI (Table 5). Age positively affected BMD in all the body parts studied (*p* < 0.05) besides L1–L4, whereas BMI was a positive predictor of all BMD values (*p* < 0.01 for leg; *p* < 0.001 for L1–L4, arm, trunk and total BMD). Among the biochemical parameters, serum testosterone was a positive factor of both the trunk and total BMD (*p* < 0.01). In contrast, BMD was negatively affected by serum OC (*p* < 0.001 for the leg; *p* < 0.01 for the trunk and total body). 

Apart from testosterone and OC, none of the biochemical parameters was a predictor of BMC/BMD values (Table 4 and Table 5). Of all the polymorphisms examined, one of the polymorphisms of gene coding the receptor of vitamin D (*FokI VDR*) and the polymorphism of gene coding the receptor of calcitonin (*CALCR*) turned out to be predictors of BMD values (Table 5). Of the three genotypes within *VDR FokI* polymorphism, GG was found to be a negative predictor of BMD at L1–L4 (*p* < 0.01), the leg, trunk and total body (*p* < 0.05, Table 5), with AG being a negative predictor in a non-significant manner. On the other hand, as indicated from ANOVA analyses, BMD at L1–L4 was significantly higher in AG TR than in AG CON (*p* = 0.01; with main effect of TR vs. CON, *p* = 0.001 and main effect of *FokI*, *p* = 0.047; Figure 5). In addition to this, the only main effect of TR vs. CON (without any *FokI* effect) was seen in the BMD of the trunk (*p* < 0.05), while no significant effects were found in the BMD of the leg or total body.

As for *CALCR* polymorphism, the AG genotype was a positive predictor of BMD value, but only in the case of the arm (*p* < 0.05, Table 5). The same was true for the GG genotype, but in a non-significant manner (Table 5). In turn, as shown in Figure 6, the arm’s BMD was found to be higher in AA TR as compared to in AA CON (*p* = 0.02; with main effects of TR vs. CON, *p* = 0.039 and the interaction of *CALCR* x TR vs. CON, *p* = 0.022).

## 3. Discussion

The most relevant finding of our study is that the common polymorphisms within the genes encoding antioxidant enzyme proteins (*SOD2* and *GPx*) are the predictors of BMC in young, healthy men. Furthermore, *SOD2, FokI VDR* and *CALCR* polymorphisms seem to modulate the associations of BMC/BMD values with training status (trained vs. untrained). 

As confirmed by our regression analyses, BMI and age seem to be positive predictors of BMC and BMD (Table 4 and Table 5), which is in line with the finding from other studies on the young, physically active, male population [4,28]. Indeed, it has been reported that, within a healthy weight range, there is a direct positive relationship between BMI and bone density [29]. In turn, positive relationships between age and bone parameters indicate that the skeleton is not fully developed. Thus, our participants were in the phase of peak bone mass development.

The main purpose of our study was to evaluate genetic factors as predictors of bone mass in young men engaged in competitive sports. We asked athletes from combat and team sports to take part in the current study, since these sports have been previously reported to have the highest positive stimulatory effects on bone formation [8]. Our results showed that the most evident positive influence of sports training on bone parameters was found at the lumbar spine (Table 1, Table 4 and Table 5), possibly due to high amounts of trabecular bone that is more metabolically active than cortical bone [29]. On the other hand, when the relationship between sports and bone parameters in other areas of the skeleton was considered, only handball proved to be a positive predictor of both BMC and BMD at almost all the sites measured, including total body (Table 4 and Table 5). Furthermore, handball, wrestling and MMA (but not soccer) were positive predictors of arm BMD, which is compatible with the nature/specificity of these sports. It confirms previous suggestions that the level of the bone adaptation through the exercise seems to be dependent on the overload, and it seems to be specific for the spots that are submitted to major stress [29].

Meanwhile, in the current study, a lack of significant influence of soccer on any BMD value was quite unexpected since the loading pattern of this sport is considered to result in high-impact forces on the bones of the lower limbs and to improve osteogenesis [28]. Instead, in our study, the only positive impact of soccer regarded BMC at the lumbar spine. The least marked effect of soccer, compared to the other disciplines studied, on bone parameters may be related to the lowest mean value of body mass in soccer as compared to other sports and those in the control group. As we aimed to evaluate all predictors of crude bone mass parameters, the analyses were not adjusted to anthropometric indices, which may be a limitation of the study. Another reason for these differences between the sports studied regarding the effect on BMD might have resulted from the different competitive levels of our athletes representing different sports, with the lowest competitive level being represented by the soccer players. However, to confirm the above-mentioned issues, further research is needed.

A study on the elderly population indicated that oxidative stress may be an independent risk factor for osteoporosis, supporting the hypothesis that links oxidative stress (resulting from a decrease in enzymatic antioxidant potential) with the etiology and physiopathology of this disease [22]. On the other hand, regular physical activity/sports training is considered as an important factor that reinforces antioxidant potential (especially enzymatic) and alleviates resting levels of oxidative stress parameters [27]. In our study, no intergroup differences were found in blood levels of enzymatic antioxidant defense and oxidative stress parameters. On the other hand, the decrease in nonenzymatic antioxidant potential (plasma TAC) found in our TR group compared to CON group was probably due to a decrease in plasma UA (Table 2) as a result of an adaptation to training, which is in line with previous findings on both endurance and speed-power athletes [30]. 

None of the parameters of prooxidant-antioxidant status was associated with bone mass parameters. On the other hand, our study is the first to find the association of bone parameters (BMC) with polymorphisms within genes encoding antioxidant enzymes in a young, healthy population.

Among the three polymorphisms studied, the relationship between *SOD2* polymorphism and BMC values was the most evident in our population, which is not surprising since SOD2 is considered as the key mitochondrial enzyme in the oxidative stress metabolism pathway [20].

So far, the only study on a European population (postmenopausal women) showed no association of SNP rs4880 of the *SOD2* gene to lumbar and femoral BMD [26]. In turn, in another cross-sectional study on Asian Indians, the *SOD2* G allele was found to be significantly more frequent in osteoporotic individuals [20].

Consistent with Botre’s findings [20], our results also suggest that the G allele of *SOD2* gene may be disadvantageous for bone health, as GG (although not significantly) and primarily AG genotypes were generated by our regression model as a negative predictor of BMC. Moreover, ANOVA analyses revealed that, in our whole population, heterozygote AG had lower BMC values than both homozygote AA (Figure 1, Figure 2, Figure 3 and Figure 4, panel A) and GG (Figure 1A). However, when BMC values (at leg, trunk and total) were analyzed separately in two groups (TR and CON), only the TR group had significant discrepancies between genotypes (Figure 2, Figure 3 and Figure 4, panel B). It may suggest that the AG genotype blunts or slows down the rate of BMC increase induced by sports training. However, it does not seem to apply to the lumbar spine, since BMC at L1–L4 was higher in the GG genotype in the TR group as compared to the same genotype of the CON group (Figure 1B), probably due to the better response of the lumbar spine to exercise as compared to other regions of the skeleton [29].

It is difficult to explain why variant G [20] or genotype AG (our study, especially in trained persons) is associated with worse bone tissue condition in comparison to the A allele/AA genotype. It has been previously reported that the G allele, in comparison with the A allele, is related to reinforcement of the enzymatic antioxidant defense as a result of more effective transport of SOD2 protein to mitochondria and an increase in SOD2 activity [24]. In line with the above, are our previous studies on healthy men who underwent chronic endurance training [31] or wrestlers [32], in which the *SOD2* GG genotype was associated with lower oxidative stress and muscle damage parameters as compared to two other genotypes. 

The explanation for a negative impact of *SOD2* GG or AG genotypes (but not the AA genotype) on bone tissue condition is possibly an imbalance between SOD and GPx activity. In the elderly Mexican population, a decrease in GPx activity, along with an increase in the SOD/GPx ratio, was seen in persons with osteoporosis [22]. As explained, greater SOD activity with respect to GPx results in an increase in H_2_O_2_ levels, which favors the differentiation of osteoblastic cells to osteoclasts and inhibits the differentiation of osteoblastic cells to osteoblasts, thus, propitiating an accentuated diminution of BMD [22]. In our study on a young population, no association was seen between BMD/BMC and antioxidant enzymes activities. However, variations in the coding region of antioxidant enzyme genes may not necessarily affect the protein’s activity but may affect its subcellular localization [33]. Therefore, it cannot be excluded that diminished BMC in our athletes carrying AG genotypes may result from some imbalance between SOD and GPx adaptation to exercise. Finally, not only the AG genotype within *SOD2* gene, but also the CT genotype within *GPx* gene, was a negative predictor of BMC in our participants (Table 4). In the study of Mlakar et al. [25] on the elderly population, the TT genotype of the *GPx* gene was related to decreased BMD of the lumbar spine, femoral neck and total hip. Furthermore, the T allele was found to be associated with reduced GPx activity in a dose-dependent manner [34]. It must be mentioned that we did not find the TT genotype in our study population, which is in line with the results of the study on other populations [35]. As we described previously [36], the lack of TT (Leu/Leu) genotype in our study population (with deviation from Hardy–Weinberg equilibrium) may point to the prevalence of the C (Pro) allele in the healthy, physically fit population. 

Among different factors, androgens take part in building the skeleton of young men and help to prevent bone loss in elderly men [37] by having antiapoptotic effects on osteocytes and osteoblasts and apoptotic effects on osteoclasts [38]. The results of our study on a young, male population confirmed the above-mentioned issue, since serum testosterone concentration was found to be a positive predictor of both trunk and total BMD.

The process of formation and resorption happens alternatively in bone development during growth and aging [2]. Serum OC is considered a specific marker of osteoblast function, as its levels have been shown to correlate with bone formation rates [39]. Furthermore, OC has been shown to be sensitive to alterations in bone metabolism due to physical exercise [40]. However, in our study, no differences in serum OC (which was at levels observed in a previous study in a young, male population [40]) were found between the CON and TR groups. Instead, our results revealed that serum OC was a negative predictor of BMD at leg, trunk and the whole body (Table 5), which was surprising in our young, healthy men. On the other hand, serum OC may be considered as the marker of overall bone turnover [39]. It has been suggested that an increase in the levels of bone turnover markers is followed by a decrease in BMD after menopause but also by an increase in BMD during puberty [41]. Therefore, in our young people in the phase of increasing their peak bone mass, an increase in serum OC with lower BMD may indicate more active bone turnover during the formation of peak BMD.

Bone turnover is mentioned to be influenced by vitamin D and calcium status. A study on a large, representative sample of healthy adolescents [42] showed that boys with high vitamin D status had significantly lower serum osteocalcin than those with low and moderate vitamin D status. 

In our study, calcium intake was at a moderate level (mean values: 822.1 ± 360.3 and 792.2 ± 436.3 mg/day for CON and TR groups, respectively; data of dietary analysis are not shown in this paper). Additionally, the mean values of 25-OH D concentration in serum were within the reference ranges. Moreover, no relationship was found between areal BMD and serum Ca or serum 25-OH D concentration. However, taking into account individual vitamin D status (serum 25-OH D values) among the whole study population, vitamin D deficiency/insufficiency (below 30 ng/mL [38]) concerned 21% and 33% of the participants from the TR and CON groups, respectively. In addition, only in the subgroup with 25-OH D < 30 ng/mL, we observed a reverse correlation between serum OC and 25-OH D, which confirms previous findings [42]. 

Serum 25-OH D was not a predictor of BMD values in the present study. However, it must be pointed out that the polymorphisms of the *VDR* gene are well-known, important factors affecting vitamin D bioavailability and/or calcium absorption and, as a result, bone mass [43,44]. Among *VDR* polymorphisms measured in our study, only *VDR FokI* start codon polymorphism was associated with BMD values. 

Our regression analysis proved the *FokI* genotypes to be independently related to BMD values, with the GG genotype (and the AG genotype, but not significantly) being a negative predictor of BMD at almost all sites except for the arm (Table 5). 

However, the results of previous studies are conflicting since the genotype–phenotype relationship is not uniform [45,46]. Our finding that the G allele seems to be unfavorable to bone tissue condition is in opposition to previous results on Swedish adolescent boys [46], where the FF (GG) genotype was related to higher BMD compared to ff (AA). Additionally, the A–G transition is generally mentioned as resulting in the synthesis of a smaller VDR protein with increased biological activity [47]. These discrepancies should be clarified in further studies, but one explanation could be a possible interaction of the effects of the *FokI VDR* polymorphism and physical activity/exercise training on bone metabolism [47]. In our study, the AG genotype of the TR group had higher spine BMD than the AG genotype of the CON group (Figure 5). In addition, a clear intergroup difference (but not significant) was also seen in the case of the AA genotype (Figure 5). Thus, our results indicate that genotypes considered as being less favorable for BMD [45,46] appear to be more susceptible to changes induced by sports training. 

Our findings are in line with a previous study by Laaksonen et al. [45] on Finnish adolescent boys, in which the AG genotype was associated with higher forearm BMD compared to the GG genotype, with these inter-genotype differences being significant only after adjusting for physical activity levels. In addition, in another investigation on elderly people involved in resistance training [48], the AG genotype was related to the highest responses in BMD in the femoral neck. Finally, a Brazilian study on adolescent soccer players also found higher total BMC and BMD in the AG genotype compared to the GG genotype [49]. 

In addition to *VDR FokI* polymorphism, the AluI polymorphism in *CALCR* gene was also associated with BMD values in our participants. This polymorphism (T to C substitution, which causes substitution of a leucine amino acid in 463 site with proline) has been reported to alter the secondary structure and, as a result, to affect the biological activity of the receptor of calcitonin, playing an important role in bone and calcium homeostasis [13]. As reported, a lack of proline residue could decrease the calcitonin hormone bond to the calcitonin receptor [50]. Accordingly, we found that the GG (CC) genotype (i.e., homozygous proline) of the *CALCR* gene was more frequent in the TR than the CON group (Table 3), which may suggest that this genotype is beneficial for athletes in combat and team sports. Additionally, both genotypes with the G allele (GG and AG) were positive predictors of arm BMD, although only AG reached a significant level (*p* < 0.05), probably because of the low frequency of the GG genotype in our participants.

The results of studies evaluating the association between *CALCR* polymorphism and BMD are inconsistent, with different genotypes indicated as unfavorable, i.e., GG in middle-aged and older men [14] and AA in postmenopausal women [13], while no association was found in a population of young, healthy men [15]. Rather, our results are in line with the findings of the study on postmenopausal women of Caucasian origin [51], in which significantly higher BMD at the femoral neck was seen in heterozygous subjects (AG) compared with in the homozygous leucine (AA) and proline (GG) genotypes. Additionally, a heterozygote was related to decreased fracture risk in that population [51]. However, it must be noted that, similar to our study, there were few subjects with the GG genotype. On the other hand, it has been suggested that heterozygotes are the most beneficial for bone health, as expression of both alleles of the receptor confers an advantage over homozygotes [50,51].

Although, in our study, the GG genotype of *CALCR* was overrepresented in the TR group, and the AG genotype was found to be a positive predictor of arm BMD, ANOVA indicated higher arm BMD in the AA genotype of the TR group than in the AA genotype of the CON group (Figure 6). Thus, even though the G allele seemed to be a positive predictor of arm BMD in our subjects, our results described above indicated that practicing combat (wrestling, MMA) and team sports (handball) improves arm BMD, and this improvement was more evident in the genotype with the lowest bone density (i.e., AA), as indicated in another study [13].

## 4. Material and Methods

### 4.1. Subjects

The study was carried out on 181 young male volunteers during the period of reaching peak bone mass (20–23 years) in the spring of 2018. Taking into account their physical activity level, they were divided into two groups: control (n = 87) and trained (n = 94). The control group consisted of untrained students from the Faculty of Physical Education and Health in Biała Podlaska who regularly participated in practical classes included in the study curriculum for 3 years and declared that they did not perform regular physical activity outside of the physical education classes at university (5 h per week). Exclusion criteria were as follows: a recent leg injury/stress fracture, a history of musculoskeletal diseases, current smokers, chronic steroid treatment, taking any medications and dietary supplements less than 3 months before the study. 

The trained group included men practicing different kinds of sports in local sports clubs, such as soccer (n = 44), wrestling (n = 22), handball (n = 13) and mixed martial arts (MMA, n = 15). Soccer players (representing 4th league soccer clubs) and handball players (representing 1st league handball clubs), as well as MMA fighters (at a collegiate level), were students from the Faculty of Physical Education and Health in Biała Podlaska, whereas the wrestlers (at a national level) were current or previous students from the Sports School in Radom. All these athletes declared long training experience (7.5 ± 2.5 years) and weekly training loads of 10–12 h.

All the participants provided their written, informed consent to take part in the study. The study was in compliance with the Helsinki Declaration. The protocol of the study was approved by the Local Ethics Committee at the University of Physical Education in Warsaw (no. SKE 01-24/2015).

### 4.2. BMC and BMD Measurements

The height (cm) of the athletes that participated in the study was measured with an accuracy of 1 mm by using an anthropometric set in a barefoot position with feet placed on the ground on one level, heels joined, the knees stretched and upright. The weight (kg) was measured with an electronic scale with clothes as thin as possible on the athletes with an accuracy of 100 g. 

Bone mineral content BMC (g) and bone mineral density BMD (g/cm^2^) were obtained from a whole-body scan with the use of dual-energy X-ray absorptiometry (DEXA) on a HORIZON Ci device (USA). The following parts of the body were taken into account: the lumbar spine (L_1_–L_4_), the arm and the leg (averaged values of the left and the right limb), as well as trunk and total body. For the total-body scan, participants were asked to lie in the supine position, centered within the scan field. The hands were placed on the sides of the legs in the prone position, while the legs were straight and strapped together. 

### 4.3. Blood Sampling and Biochemical Analyses

Fasting blood samples were taken from the ulnar vein in the morning (at 7:00 a.m.). All the subjects had not eaten any food for 10–12 h and had not exercised for 24 h prior to blood drawing. The samples were collected using test tubes with EDTA (for the whole blood, erythrocyte and plasma analyses, as well as DNA isolation) and without anticoagulants (for separation of serum). In order to form a blood clot, the samples were exposed at room temperature and then centrifuged (for 10 min at 3000× *g* at a temperature of 4 °C) to separate serum. Additionally, the portion of the blood with EDTA was centrifuged to separate erythrocytes and plasma. Subsequently, the erythrocytes were washed three times with a cold, isotonic saline solution. Erythrocytes, plasma, serum and whole blood were frozen and stored at −80 °C until analysis.

The measured biochemical parameters were activity of superoxide dismutase (SOD) in erythrocytes, activity of glutathione peroxidase (GPx) in whole blood and total antioxidant capacity (TAC) of plasma, as well as plasma concentration of uric acid (UA), lipid hydroperoxides (LOOHs) and phosphates (P). Serum was analyzed for the activity of alkaline phosphatase (ALP), as well the concentration of calcium (Ca), osteocalcin (OC), testosterone (T) and 25-OH vitamin D (25-OH D).

SOD and GPx activities were determined with commercially available kits (RANSOD cat. no. SD 125 and RANSEL cat. no. RS 505, respectively; Randox, Crumlin, UK). The antioxidant enzyme activities were measured at 37 °C and expressed in U/g Hb. Hemoglobin was assessed using a standard cyanmethemoglobin method with a diagnostic kit (cat. no. HG 1539; Randox, Crumlin, UK). The enzymatic activities were measured at 37 °C and expressed in U/g Hb.

The total antioxidant capacity of plasma (TAC) to scavenge ABTS radicals was measured using a chromogenic method with a commercially available kit (cat. no. NX 2332; Randox, Crumlin, UK). The antioxidant capacity of samples was expressed as millimoles per liter of Trolox equivalents (6-hydroxy-2,5,7,8-tetramethylchroman-2-carboxylic acid).

LOOHs levels were determined as described previously [52]; the assay was based on the reaction of a chromogenic reagent, N-methyl-2-phenylindole, with malondialdehyde and 4-hydroxyalkenals at 45 °C. As a result, a stable chromophore was formed with maximum absorbance at 586 nm.

Serum levels of UA (cat. no. K6581–100), Ca (cat. no. W6504–100), P (cat. no. F6516–100) and ALP activity (cat. no. F6406–075) were determined with commercially available kits (Alpha Diagnostics, Poland) using multicalibrator (cat. no. K6504–03), normal and pathological control serum (cat. nos. S6590–05 and S6591-05, respectively; Alpha Diagnostics, Poland). These biochemical parameters (as well as SOD, GPx, TAC and LOOHs) were measured spectrophotometrically at 37 °C using an automatic biochemical analyzer A15 (Bio-Systems S.A., Montcada I Reixac, Spain) by a certified laboratory diagnostician (K.J.) to minimize as much as possible the influence of inter-assay variation.

The concentration of osteocalcin, testosterone and 25-OH vitamin D was measured with the ELISA method using diagnostic sets (Cormay, Poland) at a local, certified, clinical, diagnostic laboratory. All commercially available kits listed above were IVD (in vitro diagnostic).

### 4.4. Genotyping

Genomic DNA for genotyping was isolated from peripheral venous blood using a QIAamp DNA Blood Mini Kit (Qiagen GmbH, Hilden, Germany). Concentration of DNA was determined with Picodrop microliter spectrophotometer (PicoDrop, UK). The following gene polymorphisms were genotyped with commercially available TaqMan kit (Applied Biosystems, Foster City, CA, USA): vitamin D receptor (*VDR*), i.e., *ApaI* (rs 7975232, C_28977635_10, cat. no. 4351379), *BsmI* (rs1544410, C_8716062_20, cat. no. 4351379) and *FokI* (rs 2228570, C_12060045_20, cat. no. 4351379); type 1 collagen *COLIA1* (rs 1800012, C_7477170_30, cat. no. 4351379); calcitonin receptor *CALCR* (rs 1801197, C_2541576_1, cat. no. 4351379); and antioxidant enzymes, i.e., *SOD1* A-39 C (rs2234694, C_34770911_10, cat. no. 4351379), *SOD2* Ala-9Val (rs4880, C_8709053_10, cat. no. 4351379) and *GPx* Pro198Leu (rs1050450, C_175686987_10, cat. no. 4351379). Genotyping for all gene polymorphisms was carried out by a 10 µL PCR reaction on DNA (50 ng) using a TaqMan PCR Master Mix (5 µL, Applied Biosystems) and fluorescent 5′-exonuclease TaqMan SNP assays (0.5 µL, Applied Biosystems) with FAM and VIC fluorophore-labeled probes. Real-time PCR was performed on Rotor Gene (Qiagen GmbH, Hilden, Germany) according to the following protocol recommended by the manufacturer of TaqMan Assays (Applied Biosystems): an initial 5 min at 95 °C followed by 40–45 cycles of 15 s at 95 °C, 30 s at 60 °C, 30 s at 72 °C and, finally, 8 min at 72 °C. For quality control, positive and negative controls and blinded duplicate samples were run. Sample images of TaqMan analysis results for each of the polymorphisms tested are included in the Appendix A. 

### 4.5. Statistical Analysis

Anthropometric data, biochemical parameters and the parameters of bone tissue condition (BMC and BMD at different points of body skeleton) in both groups (trained and control) were analyzed with one-way ANOVA using Statistica version 13.3 software package (StatSoft, Krakow, Poland). Moreover, relationships between biochemical parameters were analyzed on the basis of Pearson’s coefficients of linear correlation. The normal distribution of all variables was confirmed with the Shapiro–Wilk test and visual inspection (quantile distribution plots). All values were reported as mean ± standard deviation (SD). The level of statistical significance was set at *p* < 0.05.

For each SNP, deviation of the genotype frequencies from those expected under the Hardy–Weinberg equilibrium was assessed in both groups with chi-square (χ^2^) test [53]. Genotype frequencies in CON and TR groups were compared using a likelihood ratio (χ^2^ test). 

Multiple regression was the basic data analysis tool [54]. The model was built in an iterative “brute force” way using r1071 R package [55] by testing all combinations of predictors (in this case 2^18^ − 1) in order to avoid the influence of the order of substitution of predictors on the model quality. The model with the lowest Akaike information criterion (AIC) was selected.

## 5. Conclusions

Age, BMI, training status, serum testosterone and osteocalcin, as well as genetic factors, are the predictors of bone mass parameters in young, healthy men. Practicing combat sports (wrestling, MMA) and team sports (handball) is an independent positive predictor of BMD/BMC values at various skeletal sites. The main finding of this study is that, at almost all measured skeletal sites, the *SOD2* AG and *FokI VDR* GG genotypes seem to be negative predictors of BMC and BMD, respectively, while the *CALCR* AG genotype appears as a positive predictor of arm BMD values. Furthermore, these three genetic polymorphisms seem to modulate the response of bone mass parameters to sports training. Overall, at least within the *VDR FokI* and *CALCR* polymorphisms, less favorable genotypes in terms of BMD (i.e., *FokI* AG and *CALCR* AA) appear to be associated with a greater BMD response to sports training. This suggests that, in healthy men during the period of bone mass formation, sports training (combat and team sports) may attenuate the negative impact of genetic factors on bone tissue condition, possibly reducing the risk of osteoporosis in later age.

## Figures and Tables

**Figure 1 ijms-24-03373-f001:**
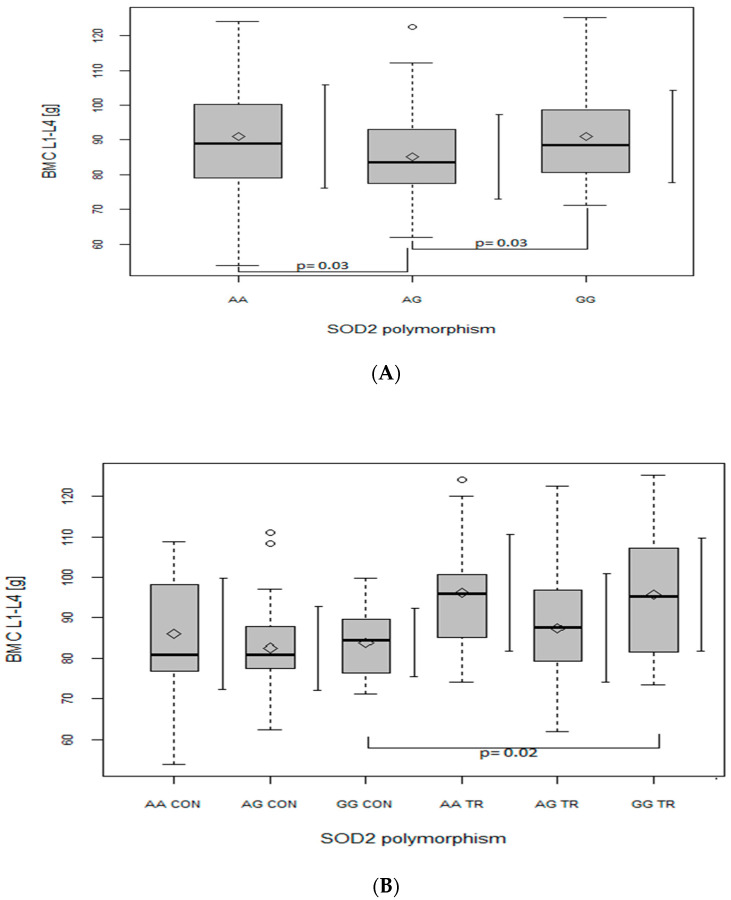
The effect of SOD2 polymorphism on bone mineral content (BMC) (g) of the lumbar spine (L1–L4). (**A**) In the whole group; main effects: SOD2 (*p* = 0.02; F = 4.23). (**B**) In control (CON) and trained (TR) group; main effects: SOD2 (*p* = 0.01; F = 4.67). TR vs. CON (*p* = 0.000000001; F = 18.1). Key: ◊—mean; ○—outliers value; boxplots—median Q1–Q4 and mean ± SD.

**Figure 2 ijms-24-03373-f002:**
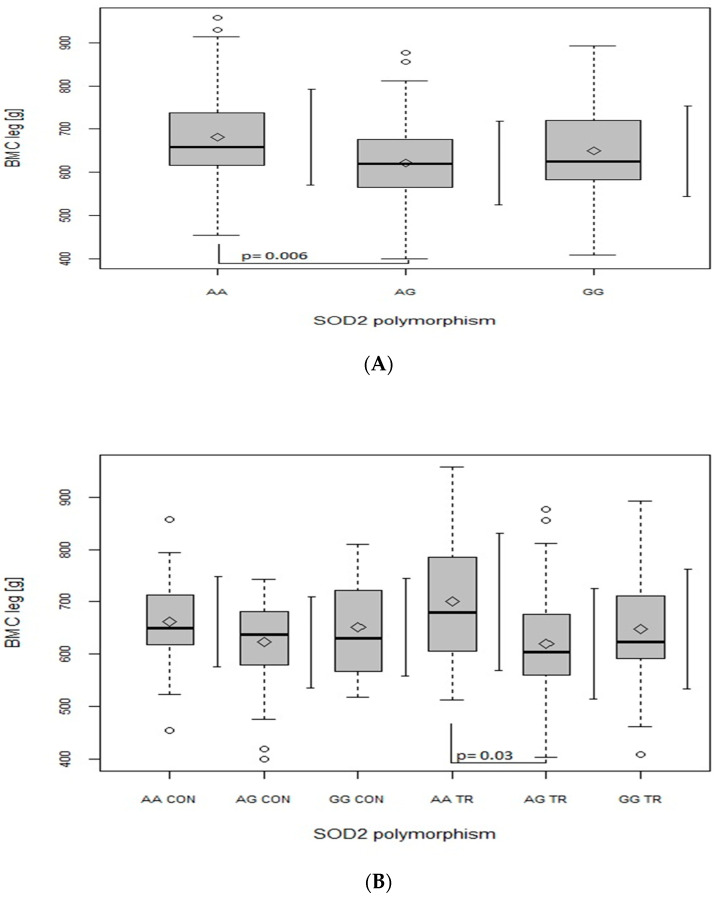
The effect of SOD2 polymorphism on bone mineral content (BMC) (g) of the leg. (**A**) In the whole group of studied participants; main effects: SOD2 (*p* = 0.008; F = 4.99). (**B**) In control (CON) and trained (TR) group; main effects: SOD2 (*p* = 0.008; F = 4.95). Key: ◊—mean; ○—outliers value; boxplots—median Q1–Q4 and mean ± SD.

**Figure 3 ijms-24-03373-f003:**
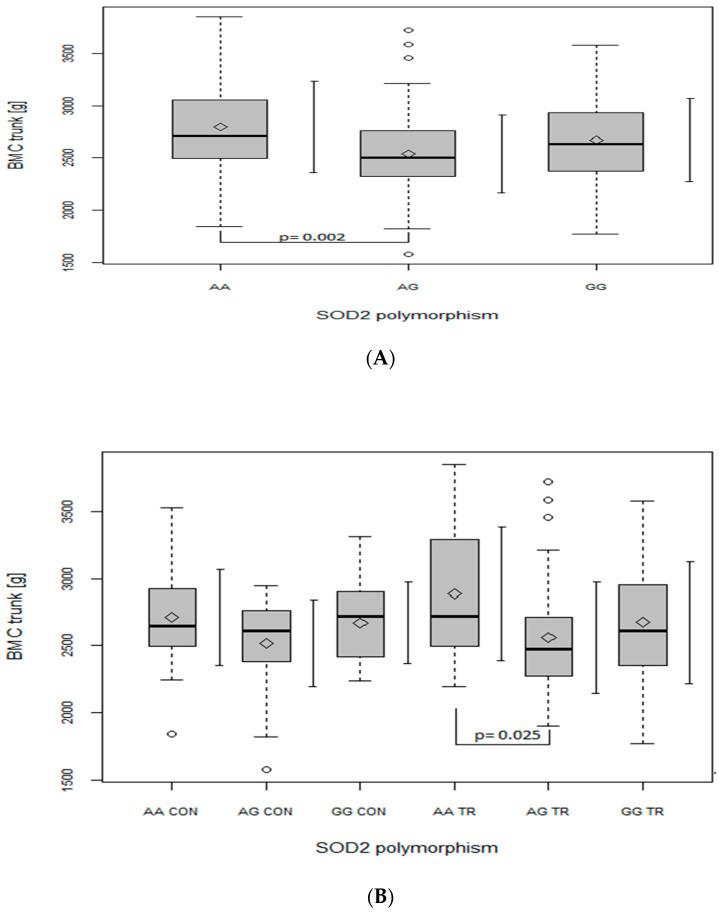
The effect of SOD2 polymorphism on bone mineral content (BMC) (g) of the trunk. (**A**) In the whole group; main effects: SOD2 (*p* = 0.002; F = 6.24). (**B**) In control (CON) and trained (TR) group; main effects: SOD2 (*p* = 0.002; F = 6.22). Key: ◊—mean; ○—outliers value; boxplots—median Q1–Q4 and mean ± SD.

**Figure 4 ijms-24-03373-f004:**
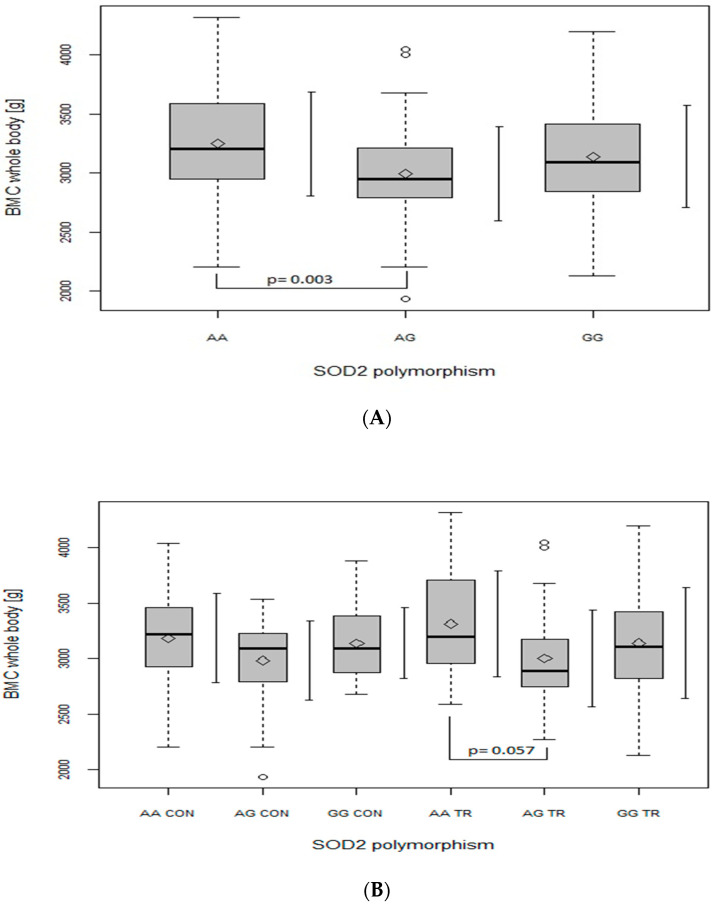
The effect of SOD2 polymorphism on bone mineral content (BMC) (g) of the whole body. (**A**): in the whole group; main effects: SOD2 (*p* = 0.004; F = 5.71). (**B**): in control (CON) and trained (TR) group; main effects: SOD2 (*p* = 0.004; F = 5.65). Key: ◊—mean; ○—outliers value; boxplots—median Q1–Q4 and mean ± SD.

**Figure 5 ijms-24-03373-f005:**
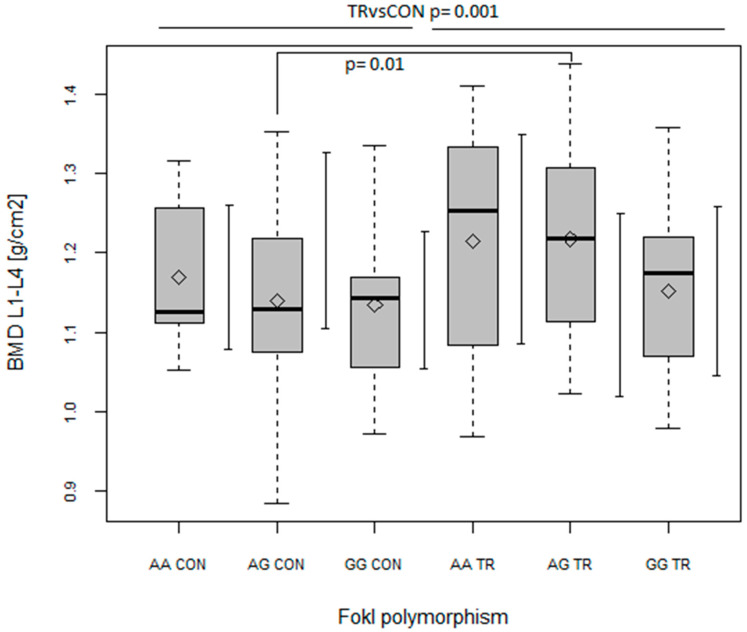
The effect of FokI polymorphism on bone mineral density (BMD) (g/cm^2^) of the lumbar spine (L1–L4) in control (CON) and trained (TR) group. Main effects: FokI (*p* = 0.047; F = 2.95). TR vs. CON (*p* = 0.001; F = 10.46). Key: ◊—mean; boxplots—median Q1–Q4 and mean ± SD.

**Figure 6 ijms-24-03373-f006:**
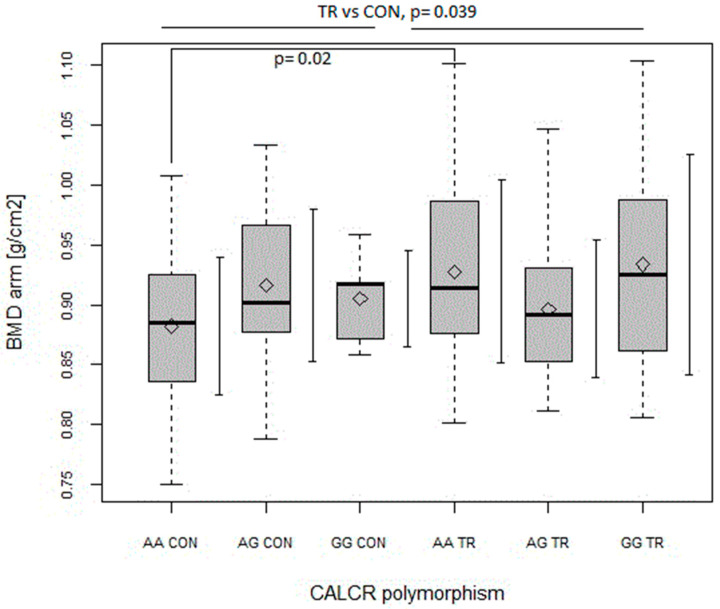
The effect of CALCR polymorphism on bone mineral density (BMD) (g/cm^2^) of the arm in control (CON) and trained (TR) group. Main effects: CALCR NS TR vs. CON (*p* = 0.039; F = 4.31). CALCR x TR vs. CON (*p* = 0.022; F = 3.88). Key: ◊—mean; boxplots—median Q1–Q4 and mean ± SD.

**Table 1 ijms-24-03373-t001:** Anthropometric data, regional and total body bone mineral content (BMC) (g) and bone mineral density (BMD) (g/cm^2^) in the control group (CON, n = 87) and in the trained group (TR, n = 94).

		CON (n = 87)	TR (n = 94)	ANOVA *p*-Value
Anthropometric data	age (years)	21.18 ± 1.75	20.96 ± 2.26	*p* = 0.48
	height (cm)	182.28 ± 5.28	180.78 ± 8.16	*p* = 0.16
	body mass (kg)	81.10 ± 9.73	80.38 ± 11.56	*p* = 0.66
	BMI (kg/m^2^)	24.39 ± 2.56	24.54 ± 2.53	*p* = 0.71
L1–L4	BMC	83.87 ± 11.04	92.20 ± 14.28	*p* = 0.00004
	BMD	1.14 ± 0.10	1.20 ± 0.12	*p* = 0.0008
arm	BMC	217.81 ± 29.50	219.73 ± 41.63	*p* = 0.73
	BMD	0.90 ± 0.06	0.92 ± 0.08	*p* = 0.030
leg	BMC	641.04 ± 89.14	648.96 ± 118.16	*p* = 0.62
	BMD	1.49 ± 0.15	1.54 ± 0.16	*p* = 0.06
trunk	BMC	2609.18 ± 337.41	2677.09 ± 463.92	*p* = 0.28
	BMD	1.21 ± 0.10	1.25 ± 0.11	*p* = 0.025
whole body	BMC	3079.59 ± 371.80	3123.20 ± 479.68	*p* = 0.51
	BMD	1.29 ± 0.10	1.31 ± 0.11	*p* = 0.075
	Z-score (min; max)	1.01 ± 0.84 (−0.7; 2.8)	1.31 ± 0.96 (−0.6; 3.6)	*p* = 0.027
	T-score (min; max)	0.93 ± 0.82 (−0.7; 2.8)	1.17 ± 0.99 (−0.7; 3.5)	*p* = 0.076

Data are mean ± SD; L1–L4—lumbar spine.

**Table 2 ijms-24-03373-t002:** Blood levels of biochemical parameters (mean ± SD) in the control group (CON, n = 87) and in the trained group (TR, n = 94).

Variables	Reference Range	CON (n = 87)	TR (n = 94)	ANOVA *p*-Value
TAC [mmol/L]	1.30–1.77	1.72 ± 0.18	1.64 ± 0.15	*p* = 0.002
GPx [U/g Hb]	27.5–73.6	60.1 ± 26.1	63.8 ± 26.0	*p* = 0.33
SOD [U/g Hb]	1102–1601	1507.4 ± 509.1	1518.7 ± 468.2	*p* = 0.88
LOOHs [µmol/L]	-	3.06 ± 1.65	2.72 ± 1.79	*p* = 0.17
ALP [U/L]	40–129	82.4 ± 23.5	76.9 ± 22.0	*p* = 0.21
Ca [mg/dL]	8.60–10.30	8.05 ± 0.89	8.31 ± 0.53	*p* = 0.11
UA [mg/dL]	3.40–7.00	5.55 ± 1.10	4.94 ± 0.96	*p* = 0.0001
P [mg/dL]	2.60–4.50	3.67 ± 0.55	3.66 ± 0.82	*p* = 0.92
T [ng/mL]	2.2–10.5	6.10 ± 1.45	5.86 ± 1.38	*p* = 0.25
OC [ng/mL]	<5.0	26.31 ± 10.01	24.26 ± 10.00	*p* = 0.18
25-OH D [ng/mL]	30–50	40.29 ± 28.11	36.18 ± 10.69	*p* = 0.20

TAC—total antioxidant capacity, SOD—superoxide dismutase, GPx—glutathione peroxidase, LOOH—lipid hydroperoxides, ALP—alkaline phosphatase, Ca—calcium, UA—uric acid, P—phosphates, T—testosterone, OC—osteocalcin, 25-OH D—25-OH vitamin D.

**Table 3 ijms-24-03373-t003:** The distribution of genotypes for gene polymorphisms in the control group (CON) and in the trained group (TR) (n (%)).

Polymorphism	CON (n = 87)	TR (n = 94)	TR vs. CON
SOD1 A-39 C			χ^2^ = 0.18 *p* = 0.67
AA	78 (89.7)	87 (92.5)	
AC	9 (10.3)	7 (7.5)	
CC	0	0	
HWE	*p* = 0.58	*p* = 0.30	
SOD2 Ala-9 Val			χ^2^ = 1.24 *p* = 0.54
AA	26 (29.9)	23 (24.5)	
AG	40 (46.0)	42 (44.7)	
GG	21 (24.1)	29 (30.8)	
HWE	*p* = 0.57	*p* = 0.40	
GPx Pro198Leu			χ^2^ = 0.69 *p* = 0.40
CC	39 (44.8)	49 (52.1)	
CT	48 (55.2)	45 (47.9)	
TT	0	0	
HWE	*p* = 0.0008	*p* = 0.005	
VDR ApaI			χ^2^ = 1.36 *p* = 0.51
AA	26 (29.9)	23 (24.5)	
AC	37 (42.5)	48 (51.0)	
CC	24 (27.6)	23 (24.5)	
HWE	*p* = 0.22	*p* = 0.94	
VDR BsmI			χ^2^ = 2.21 *p* = 0.33
AA	16 (18.4)	10 (10.7)	
AG	35 (40.2)	41 (43.6)	
GG	36 (41.4)	43 (45.7)	
HWE	*p* = 0.21	*p* = 0.88	
VDR FokI			χ^2^ = 1.45 *p* = 0.48
AA	15 (17.2)	20 (21.3)	
AG	44 (50.6)	51 (54.2)	
GG	28 (32.2)	23 (24.5)	
HWE	*p* = 0.87	*p* = 0.49	
CALCR			χ^2^ = 11.57 *p* = 0.003
AA	44 (50.6)	58 (61.7)	
AG	38 (43.6)	21 (22.3)	
GG	5 (5.8)	15 (16.0)	
HWE	*p* = 0.51	*p* = 0.00006	
COLIA1			χ^2^ = 3.68 *p* = 0.16
AA	2 (2.3)	8 (8.5)	
AC	22 (25.3)	19 (20.2)	
CC	63 (72.4)	67 (71.3)	
HWE	*p* = 0.74	*p* = 0.52	

HWE—Hardy–Weinberg equilibrium. χ^2^ chi-square (between groups TR vs. CON), *p*-values.

**Table 4 ijms-24-03373-t004:** Regression models of BMC.

	Response Variables	BMC L1–L4 (Std.Err.)	BMC Arm (Std.Err.)	BMC Leg (Std.Err.)	BMC Trunk (Std.Err.)	BMC Total (Std.Err.)
Predictors						
Age		1.561 **	2.860 *			
		(0.469)	(1.309)			
BMI			5.864 ***	12.547 ***	51.801 ***	60.093 ***
			(1.000)	(2.969)	(11.369)	(12.106)
GPx CT		−3.727 *				
		(1.831)				
SOD2 AG		−5.299 *		−36.960 *	−153.853 *	−144.693 *
		(2.208)		(16.191)	(63.413)	(67.997)
SOD2 GG		−0.746		−8.323	−57.214	−23.763
		(2.456)		(18.066)	(70.974)	(75.905)
MMA		6.897 *				
		(3.270)				
Soccer		5.300 *				
		(2.404)				
Handball		19.239 ***	53.485 ***	142.667 ***	647.764 ***	555.423 ***
		(3.914)	(10.099)	(28.281)	(113.529)	(117.715)
Wrestling		14.187 ***				
		(2.883)				
R^2^		0.310	0.456	0.386	0.381	0.354
Adjusted R^2^		0.271	0.422	0.351	0.346	0.322
RSE		11.522	29.223	84.699	332.110	356.180
F Statistic		7.927 ***	13.169 ***	11.01 6***	10.806 ***	10.909 ***

Std.Err.: standard error; RSE: residual standard error; MMA: mixed martial arts. * *p* < 0.05; ** *p* < 0.01; *** *p* < 0.001.

**Table 5 ijms-24-03373-t005:** Regression models of selected variables as predictors of BMD.

	Response Variables	BMD L1–L4 (Std.Err.)	BMD Arm (Std.Err.)	BMD Leg (Std.Err.)	BMD Trunk (Std.Err.)	BMD Total (Std.Err.)
Predictors						
TR vs. CON		0.095 ***	0.045 **			
		(0.025)	(0.016)			
Age			0.006 *	0.012 *	0.008 *	0.009 *
			(0.002)	(0.006)	(0.004)	(0.004)
BMI		0.011 ***	0.011 ***	0.014 **	0.011 ***	0.012 ***
		(0.003)	(0.002)	(0.005)	(0.003)	(0.003)
FokI GG		−0.060 **		−0.065 *	−0.052 *	−0.051 *
		(0.022)		(0.031)	(0.021)	(0.021)
FokI AG		−0.020		−0.037	−0.026	−0.026
		(0.020)		(0.028)	(0.019)	(0.019)
T					0.012 **	0.011 **
					(0.005)	(0.005)
CALCR AG			0.020 *			
			(0.009)			
CALCR GG			0.009			
			(0.013)			
OC				−0.004 ***	−0.002 **	−0.002 **
				(0.001)	(0.001)	(0.001)
MMA			0.034*			
			(0.015)			
Soccer						
Handball			0.091 ***	0.194 ***	0.146 ***	0.121 ***
			(0.018)	(0.045)	(0.030)	(0.030)
Wrestling			0.054 ***		0.046*	
			(0.014)		(0.024)	
R^2^		0.326	0.470	0.330	0.351	0.318
Adjusted R^2^		0.283	0.440	0.288	0.309	0.275
RSE		0.097	0.053	0.133	0.089	0.089
F Statistic		7.632 ***	15.575 ***	7.740 ***	8.479 ***	7.333 ***

Std.Err.: standard error; RSE: residual standard error; MMA: mixed martial arts. * *p* < 0.05; ** *p* < 0.01; *** *p* < 0.001.

## Data Availability

The data presented in this study are available on request from the corresponding author E.J.

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
