# Peer review of "Polymorphisms in Genes Encoding VDR, CALCR and Antioxidant Enzymes as Predictors of Bone Tissue Condition in Young, Healthy Men"

_ijms, 2023, doi:10.3390/ijms24043373_

Round 1
Reviewer 1 Report
The authors of the manuscript should correct many parts of the article:
1) names of genes and polimorfphisms, such as FokI, should always be in italics, throughout the whole manuscript
2) the abstract - lacks a more specific indication which exact genotypes of the studied polymorphisms influenced on obtained result; no information whether they are positive or negative predictors
3) page 3 lines 97-103 should be moved to the materials and methods section
4) the Introduction section is too long, it should be shortened and, above all, it should justify the topic and the selection of polymorphisms and genes for analysis, part of this section should be moved to the discussion section; rs numbers should be added to the descriptions of polymorphisms of the studied genes in parentheses
5) in the Materials and Methods - Subjects section, information about period of time when material was collected - it should be added
6) in the opinion of the reviewer, the study lacks a control group, because the CON group is still students of Physical Education, so they are physically active people, which is admitted by the authors of the article themselves in the discussion, so the study should include a control group and those two groups that already examined, also these groups should be named differently, e.g. athletics and non-athletics (such names are also mentioned by the authors themselves in the manuscript page 8 line 277)
7) page 5, lines 147-167, the authors should include the exact names of the reagent kits with information whether they were ROU or IVD kits, and not only the names of the manufacturers, additionally, the names of the apparatus on which these determinations were made should appear; in addition, this fragment of the text lacks a concise description of the principle of the test method and information whether positive or negative controls, standard curves, etc. were used in the analysis
8) the description of genotyping is unclear and incomplete (there is no complete information as to the specific names / numbers of TagMan tests for a given polymorphisms and the reaction mixture used; were any controls used in this part? what was it? in supplementary materials, it would be useful to include sample photos of the TaqMan analyzes results for each of the tested polymorphisms
9) was the anneling temperature the same for each of the tested polymorphisms? - it seems rather impossible; if they were different, it should be described in the manuscript
10) in the case of CALCR and GPx, the Hardy-Weinberg equilibrium was not complied with, which may indicate an incorrect selection of the sample in relation to the population or an error in the TaqMan method, therefore, the results obtained for these genes should be confirmed by another method, preferably by sequencing
11) both in the results and in the discussion, the authors should give the exact value of p
12) the authors in the results section do not accurately describe the results and the examined relationship between the frequency of occurrence of genotypes of all polymorphisms and genes between the CON and TR groups
13) apart from regression, a statistical analysis of the genotype of the tested polymorphisms vs. anthropometric features, BMC, BMD and bicochemical results for all tested genes should be performed
14) in Table 2, a column with reference values for a given parameter should be added (taking into account the selected test method, reagent kit; at least for IVD kits)
15) why not all genotypes for all tested polymorphisms (all genes) are included in Table 4, only selected ones)? should be for every polymorphism
16) the manuscript lacks figures and Table 5, which are described in the text of the manuscript in the results and discussion section, so it is impossible to relate the information from the text to the data presented on them
17) the discussion in the paper is too long, in some fragments it is a repetition of the description of the results; the discussion should be condensed and lead to conclusions from the obtained results, additionally, the authors mostly do not attempt to interpret/explain how a given variant/genotype of a specific polymorphism affects the encoded protein and thus BMC and BMD
18) p. 8 lines 274-275 - positive or negative predictor?
19) page 8 line 278 what BMI value, what age
20) page 10 line 330 how does this affect the results of research?
21) p.11 line 345 how/why does AG affect protein and then BMC?
22) in the conclusions, the authors should refer to the specific genotypes of the studied polymorphisms and should also refer to how the obtained results may affect, for example, the health of the subjects in the future (e.g. the risk of osteoporosis)
Author Response
We are extremely grateful for all critical remarks concerning our work indicating your very profound analysis of our manuscript.
1) names of genes and polimorfphisms, such as FokI, should always be in italics, throughout the whole manuscript
Response: According to suggestion, we have changed names of genes and polymorphisms (in italics).
2) the abstract - lacks a more specific indication which exact genotypes of the studied polymorphisms influenced on obtained result; no information whether they are positive or negative predictors
Response: Thank you for this remark. Such a concise form of the abstract is due to the limit of 200 words (according to the instructions for authors). The current version of the abstract is rewritten, taking into account the reviewer's rightful suggestions. Unfortunately, in the new form it is a little too elaborate, but there is no way to include all the content on the relationship between genotypes and the results obtained while maintaining 200 words.
3) page 3 lines 97-103 should be moved to the materials and methods section
4) the Introduction section is too long, it should be shortened and, above all, it should justify the topic and the selection of polymorphisms and genes for analysis, part of this section should be moved to the discussion section; rs numbers should be added to the descriptions of polymorphisms of the studied genes in parentheses
Response to comments 3-4: We admit that the introduction is rather long, which is due to the need to confirm the relevance of the topic undertaken and the choice of polymorphisms studied, as well as to present the current state of knowledge on the topic under study. Taking into account the reviewer's suggestion, we have shortened the introduction, and added the rs numbers of polymorphisms. We decided to leave the purpose of the study here, as it is an integral part of the introduction.
5) in the Materials and Methods - Subjects section, information about period of time when material was collected - it should be added
7) page 5, lines 147-167, the authors should include the exact names of the reagent kits with information whether they were ROU or IVD kits, and not only the names of the manufacturers, additionally, the names of the apparatus on which these determinations were made should appear; in addition, this fragment of the text lacks a concise description of the principle of the test method and information whether positive or negative controls, standard curves, etc. were used in the analysis
Response to comments 5 and 7: All this information has been added.
6) in the opinion of the reviewer, the study lacks a control group, because the CON group is still students of Physical Education, so they are physically active people, which is admitted by the authors of the article themselves in the discussion, so the study should include a control group and those two groups that already examined, also these groups should be named differently, e.g. athletics and non-athletics (such names are also mentioned by the authors themselves in the manuscript page 8 line 277)
The intention of the authors of this study was to assess potent predictors of BMC/BMD in the group of young healthy men during the period of reaching peak bone mass. One of the factors considered was sports training at the competitive level. The group of trained individuals-athletes consisted mostly of physical education students, so we decided to involve as a comparison group (i.e., control) physical education students who are untrained, i.e., who do not participate in any regular sports training, but only attend classes included in the study program (moderate physical activity level).
According to the definition, control group (in an experiment or clinical trial) is a group of subjects closely resembling the treatment group in many demographic variables but not receiving the active medication or factor under study (i.e. regular sports training at competitive level in our study) and thereby serving as a comparison group when treatment results are evaluated.
We agree with the reviewer that the introduction of an additional group beyond the physical education students would perhaps give additional information, presumably exacerbating the differences from the trained group. However, separating the group outside of physical education students could introduce additional variables that differentiate the groups. The fact that these would be untrained individuals would remain unchanged, but their level of physical activity would not necessarily be lower at all due to possible work-related physical activity or daily activities. In conclusion, we leave the current division of the groups (names) as the division into athletes -non-athletes could be misleading- it is not being an athlete (success achieved), but regular long-term sports training that is considered as a factor potentially affecting bone metabolism. In addition, the term "non-athletes" would not preclude these people from participating in regular training, such as at the recreational level, therefore, we have decided to avoid using the term athletes -non-athletes in the revised manuscript. Nevertheless, thank you for this valuable comment, and we are considering the possibility of introducing such an additional group in our next studies.
8) the description of genotyping is unclear and incomplete (there is no complete information as to the specific names / numbers of TagMan tests for a given polymorphisms and the reaction mixture used; were any controls used in this part? what was it? in supplementary materials, it would be useful to include sample photos of the TaqMan analyzes results for each of the tested polymorphisms
9) was the anneling temperature the same for each of the tested polymorphisms? - it seems rather impossible; if they were different, it should be described in the manuscript
Response to comments 8-9: The description of genotyping has been completed (with specific TagMan assay names/numbers, reaction mixture used and controls). As suggested, we have added sample images of TaqMan analysis results for each of the polymorphisms tested in the supplementary materials.
Genotyping conditions followed the manufacturer's recommendations for SNP genotyping analysis using TaqMan Assays (Applied Biosystems). We present additional confirmation these conditions from publications on genotyping with Taqman (list below updated with latest works), where the annealing (and often elongation) temperature is 60 degrees C.
- VDR ApaI
Ansari MGA et al. Vitamin D Receptor Gene Variants Susceptible to Osteoporosis in Arab Post-Menopausal Women. Current Issues in Molecular Biology. 2021; 43(3):1325-1334. https://doi.org/10.3390/cimb43030094
Vranić V et al. Vitamin D receptor gene variants contribute to hip and knee osteoarthritis susceptibility. Arch Biol Sci 2021;73(2):247-55. Available from: https://serbiosoc.org.rs/arch/index.php/abs/article/view/6442
-VDR BsmI
Bermúdez-Morales VH et al. Vitamin D receptor gene polymorphisms are associated with multiple sclerosis in Mexican adults. J Neuroimmunol. 2017;306:20-24. doi:10.1016/j.jneuroim.2017.01.00
Pedrera-Canal M et al. Lack of Influence of Vitamin D Receptor BsmI (rs1544410) Polymorphism on the Rate of Bone Loss in a Cohort of Postmenopausal Spanish Women Affected by Osteoporosis and Followed for Five Years. PLoS One. 2015;10(9):e0138606. Published 2015 Sep 22.
Marozik P, Rudenka A, Kobets K, Rudenka E. Vitamin D Status, Bone Mineral Density, and VDR Gene Polymorphism in a Cohort of Belarusian Postmenopausal Women. Nutrients. 2021; 13(3):837. https://doi.org/10.3390/nu13030837
-VDR FokI
Osman E et al. Frequency of rs731236 (Taql), rs2228570 (Fok1) of Vitamin-D Receptor (VDR) gene in Emirati healthy population. Meta Gene. 2015;6:49-52. Published 2015 Sep 15. doi:10.1016/j.mgene.2015.09.001
-CALCR
Suzuki A et al. Large-scale investigation of genomic markers for severe periodontitis. Odontology. 2004;92(1):43-47. doi:10.1007/s10266-004-0035-4
Ali FT et al. Association of TRPV5, CASR, and CALCR genetic variants with kidney stone disease susceptibility in Egyptians through main effects and gene-gene interactions. Urolithiasis. 2022;50(6):701-710. doi:10.1007/s00240-022-01360-z
-Col1A1
Stepien-Slodkowska, M., Ficek, K., Zietek, P., Kaczmarczyk, M., Lubkowska, W., Szark-Eckardt, M., & Cieszczyk, P. (2017). Is the combination of COL1A1 gene polymorphisms a marker of injury risk. J Sport Rehabil, 26(3), 234-238.
-SOD1, SOD2, GPx
Dragicevic B et al. Association of SOD2 (rs4880) and GPX1 (rs1050450) Gene Polymorphisms with Risk of Balkan Endemic Nephropathy and its Related Tumors. Medicina. 2019; 55(8):435. https://doi.org/10.3390/medicina55080435
Gusti AMT, Qusti SY, Alshammari EM, Toraih EA, Fawzy MS. Antioxidants-Related Superoxide Dismutase (SOD), Catalase (CAT), Glutathione Peroxidase (GPX), Glutathione-S-Transferase (GST), and Nitric Oxide Synthase (NOS) Gene Variants Analysis in an Obese Population: A Preliminary Case-Control Study. Antioxidants. 2021; 10(4):595. https://doi.org/10.3390/antiox10040595
10) in the case of CALCR and GPx, the Hardy-Weinberg equilibrium was not complied with, which may indicate an incorrect selection of the sample in relation to the population or an error in the TaqMan method, therefore, the results obtained for these genes should be confirmed by another method, preferably by sequencing
Response: Regarding the CALCR polymorphism, the lack of compliance with Hardy-Weinberg equilibrium applies only to the trained group, which may mean that a specific genotype/allel is favorable in the study group (in this case, the GG genotype is more common in the trained than in the control group, which may indicate the predominance of this genotype in mixed sports, such as combat sports and team sports). We agree that in the case of GPx, the Hardy-Weinberg equilibrium was not observed in both groups, which could be indicative of improper sampling relative to the population. However, the genotype we did not observe in our study population i.e. TT (Leu/Leu) is considered, according to reports by other authors, to be unfavorable from the point of view of resistance to oxidative stress, also associated with an increased risk of diseases with free radical origin. It appears that the TT genotype being a rare genotype in the general population, does not necessarily occur in the subpopulation of healthy individuals (recruited to the study through additional inclusion/exclusion criteria), and its absence has also been observed by other authors, as indicated in the manuscript. It must be emphasized that the absence of the TT genotype also concerned our previous studies involving physical education students, and possible reasons for this were described in our previous publication (Kotowska J, Jówko E. Effect of Gene Polymorphisms in Antioxidant Enzymes on Oxidative-Antioxidative Status in Young Men. Polish Journal of Sport and Tourism. 2020;27(4): 7-13).
As we explained, the lack of TT (Leu/Leu) genotype in our study population may point to the prevalence of C (Pro) allele in healthy physically fit population. We have added this explanation and citation in the revised manuscript (lines 437-439).
Of course, another reason for the lack of TT genotype in our study could be errors in genotyping, but when performing analyses in our previous projects (concerning the determination of GPx, SOD2, SOD1 polymorphisms) we verified some samples by determining by other methods (sequencing and RFLP- restriction fragment length polymorphism) and the results of genotyping by different methods were consistent (verifications by sequencing concerning SOD2 mentioned in our previous papers, cited in the current manuscript as 35, 36).
11) both in the results and in the discussion, the authors should give the exact value of p
Response: In the results section, exact p-values have been added wherever possible. Only in the case of BMC and BMD predictors (Tables 4 and 5 and their description) p-values were given as p< 0.05; p<0.01 or p<0.001, as they were generated from the statistical program in this form. In the Discussion section, we have omitted to provide a p value so as not to duplicate the description of the results.
12) the authors in the results section do not accurately describe the results and the examined relationship between the frequency of occurrence of genotypes of all polymorphisms and genes between the CON and TR groups
Response: The relationship between the CON and TR groups in genotype frequencies of all polymorphisms, shown in Table 3, was described in lines 221-223 of the previous version of the manuscript. As described, except for CALCR, there were no differences in genotype frequencies between the CON and TR groups
13) apart from regression, a statistical analysis of the genotype of the tested polymorphisms vs. anthropometric features, BMC, BMD and bicochemical results for all tested genes should be performed
Response: Using the R statistical program described in the methods, in addition to regression and predictors of BMC and BMD, it simultaneously allowed ANOVA (factorial) analyses (with post-hoc) to be performed to assess differences in BMC/ BMD between the genotypes and groups studied, with main effects of polymorphism, group (CON vs. TR), and interaction of polymorphism and group. Significant differences are shown in the figures and described in the results. No such relationships were observed between groups and genotypes in anthropometric and biochemical parameters. Given the reviewer's instructions, we once again analyzed (by factorial ANOVA, using Statistica software; dependent factor: biochemical parameter studied, independent factors/predictors: polymorphism/genotypes and group) all results for biochemical parameters in all polymorphisms and groups studied. No significant differences were observed between genotypes (applicable to all polymorphisms studied) and groups in any of the biochemical parameters studied. The only significant main effects were for the group effect (CON vs TR), in all polymorphisms for uric acid, as well as TAC. Differences in these parameters between groups (TR and CON) are already shown in Table 2.
14) in Table 2, a column with reference values for a given parameter should be added (taking into account the selected test method, reagent kit; at least for IVD kits)
Response: A reference range (as specified by the reagent manufacturer) for all parameters measured with IVD kits has been added to Table 2.
15) why not all genotypes for all tested polymorphisms (all genes) are included in Table 4, only selected ones)? should be for every polymorphism
Response: Tables 4 and 5 include all major predictors of BMC and BMD, respectively, which were generated by the statistical program. In Table 4, we omitted only GG SOD2 (which was indicated as a predictor of BMC, at these skeletal sites as AG SOD2, but not significant, as we mentioned in the text of our previous version of manuscript: lines 332-334). In Table 5, the system-generated FokI AG and CALCR GG were still indicated as non-significant predictors of BMD (CALCR GG mentioned in the text of our previous version of manuscript: lines 537-538). Following the reviewer's suggestion, we have inserted them into the tables and in text of the revised manuscript (lines 281-282; 314-315; 321-323; 396; 483 ). Other parameters and polymorphisms were not extracted by the system. As described in the methods, the multiple regression model was built in an iterative "brute force" manner, meaning that the program only provides the final model with the lowest AIC value (giving the most important predictors).
16) the manuscript lacks figures and Table 5, which are described in the text of the manuscript in the results and discussion section, so it is impossible to relate the information from the text to the data presented on them
Response: At the stage of sending the manuscript, it was not possible to attach a separate file with the tables, so the tables were placed in the main document, while the figures were attached in a separate file, but for some unknown reason they were not included in the final materials, for which I sincerely apologize. The file with the figures with the original submission date has now been included. In the attached manuscript in Word, all the tables are complete (each on a separate page), while when creating the PDF, tables 4 and 5 were split. In the revised manuscript, everything is corrected.
17) the discussion in the paper is too long, in some fragments it is a repetition of the description of the results; the discussion should be condensed and lead to conclusions from the obtained results, additionally, the authors mostly do not attempt to interpret/explain how a given variant/genotype of a specific polymorphism affects the encoded protein and thus BMC and BMD
Response: The overly extended discussion is due to the large number of factors analyzed and data we wanted to clarify. As for the interpretation/explanation of how a particular variant/genotype of a particular polymorphism affects the encoded protein and thus BMC and BMD, we tried to describe this in a previous version of the manuscript ( lines 349-377; 433-449; 476-479. This must be emphasized that it is difficult to explain the cause-and-effect relationship between a given genotype and protein and its effect on BMD/BMC. Especially given the contradictory results observed in different populations, which, as has been pointed out in the discussion, may be due to the interaction of genotype with other factors (diet, physical activity).
As suggested, we have shortened the discussion (as much as it is possible) and highlighted sections explaining the possible impact of SOD and GPx polymorphisms on BMC, as well as CALCR and VDR FokI on BMD.
18) p. 8 lines 274-275 - positive or negative predictor?
Response: This is general information about the fact that genes encoding SOD2 and GPx proteins are predictors of BMC in the study population. The exact information about which genotypes are positive or negative predictors is described in the following discussion.
19) page 8 line 278 what BMI value, what age
Response: I do not understand this question. “BMI and age seem to be positive predictors of BMC and BMD” means that during the period of bone mass formation, the higher the age and higher the BMI (which is within the range of normal BMI in our study population: 24.4 ± 2.6 for CON and 24.45± 2.5 for TR group) the higher the BMC/BMD values.
20) page 10 line 330 how does this affect the results of research?
21) p.11 line 345 how/why does AG affect protein and then BMC?
Response to comments 20-21: Overall, our results indicate that the G allele, especially the AG genotype (significantly, to a lesser extent the GG genotype - but not significantly, perhaps due to lower abundance) is a negative predictor of BMC. According to the literature, the G allele, compared to the A allele, is associated with more efficient transport of SOD2 protein into cell mitochondria and greater SOD2 activity in cell mitochondria, while increasing the amount of the product of superoxide anion radical dismutation (a reaction catalyzed by SOD2), namely hydrogen peroxide, which can adversely affect bone metabolism. As also shown in the current study, regular training increases BMC/BMD in young individuals (significantly higher BMC and BMD values were found in the TR group compared to CON). At the same time, as shown in Figures 1-4, the increase in BMC was most marked in the AA genotype (with the exception of L1-L4, BMC at other skeletal sites was significantly higher in the AA genotype of TR group than the AG genotype of the TR group). Hence, the finding that the AG genotype slows down the rate of BMC growth induced by sports training. An explanation could be exercise-induced adaptation - perhaps an increase in SOD2 activity in the mitochondria of cells, without parallel adaptation of GPx ( that neutralizes hydrogen peroxide), which may be less beneficial for bone formation processes. These explanations were partially addressed in the discussion.
22) in the conclusions, the authors should refer to the specific genotypes of the studied polymorphisms and should also refer to how the obtained results may affect, for example, the health of the subjects in the future (e.g. the risk of osteoporosis)
Response: In accordance with the reviewer's suggestions, the conclusions have been revised highlighting specific application aspects of the work and the results obtained.

Reviewer 2 Report
The present manuscript entitled, Polymorphisms in genes encoding VDR, CALCR, and antioxidant enzymes as predictors of bone tissue condition in young healthy men, would be a nice good research study in the field of bone biology.
The abstract is good.
The introduction is well written with a clear background, aim, etc.
Materials and methods are okay.
Result section is prepared with all tables, no graphs, and figures. Please replace some of the tables with graphs if possible to make them easy for the readers.
All tables are at the bottom of the manuscript. I think they should be after their first mention in the text.
Note: In the text, there are several figures cited frequently, but there was no Figure in the whole manuscript! Please include them if they really exist.
Author Response
We would like to express our gratitude for comments concerning our manuscript.
The present manuscript entitled, Polymorphisms in genes encoding VDR, CALCR, and antioxidant enzymes as predictors of bone tissue condition in young healthy men, would be a nice good research study in the field of bone biology.
The abstract is good.
The introduction is well written with a clear background, aim, etc.
Materials and methods are okay.
Result section is prepared with all tables, no graphs, and figures. Please replace some of the tables with graphs if possible to make them easy for the readers.
All tables are at the bottom of the manuscript. I think they should be after their first mention in the text.
Response: Tables were originally to be sent in a separate file, However, since it was not possible to include a separate file with tables at the stage of sending the manuscript, the tables were promptly placed at the end of the main document. We have left the tables at the end of the manuscript allowing the editors to decide where to put them.
Note: In the text, there are several figures cited frequently, but there was no Figure in the whole manuscript! Please include them if they really exist.
Response: I cannot explain why the figure file is not included with the manuscript. It is not excluded that I made a mistake during upload, for which I apologize. The file with the figures with the original submission date has now been included.
Round 2
Reviewer 1 Report
The authors of the manuscript made the changes indicated in the review and answered the reviewer's questions in detail.
Reviewer 2 Report
The authors improved the manuscript. I would suggest accepting this version and publishing it after confirming all of the tables and figures cited after their first mention.